# *Bombus terrestris* Prefer Mixed-Pollen Diets for a Better Colony Performance: A Laboratory Study

**DOI:** 10.3390/insects15040285

**Published:** 2024-04-17

**Authors:** Ziyu Zhou, Hong Zhang, Shibonage K. Mashilingi, Chunting Jie, Baodi Guo, Yi Guo, Xiao Hu, Shahid Iqbal, Bingshuai Wei, Yanjie Liu, Jiandong An

**Affiliations:** 1State Key Laboratory of Resource Insects, Key Laboratory of Insect-Pollinator Biology of Ministry of Agriculture and Rural Affairs, Institute of Apicultural Research, Chinese Academy of Agricultural Sciences, Beijing 100193, China; angela99517zzy@163.com (Z.Z.); zhanghong@caas.cn (H.Z.); kulindwasm@gmail.com (S.K.M.); jct970115@163.com (C.J.); baodi_guo@163.com (B.G.); guoyi63671@163.com (Y.G.); xiaohu0607@yeah.net (X.H.); shahidiqbal525592@gmail.com (S.I.); w3295780874@163.com (B.W.); 2Department of Crop Sciences and Beekeeping Technology, College of Agriculture and Food Technology, University of Dar es Salaam, P.O.BOX 35091, Dar es Salaam, Tanzania

**Keywords:** pollinator, bumblebee, food preference, pollen, nutrition, colony development

## Abstract

**Simple Summary:**

Bumblebees are important pollinators for many plants, and plant pollen is the most important food for bumblebees in return. However, there are significant differences in the nutrition compositions of pollens from different plants, and it is still not known whether bumblebees determine their feeding behavior according to the nutritional content. In the current study, we compared the feeding responses of the managed *Bombus terrestris* on four commercial pollens and their effect on colony development under laboratory conditions. The results show that *B*. *terrestris* preferred wild apricot pollen, followed by oilseed rape, buckwheat, and, lastly, sunflower pollens. The number, body weight, and size of the bumblebees in colonies fed with a diet combination of four pollens were significantly higher than those of the bumblebees in colonies fed any single-pollen diet; even buckwheat and sunflower fed alone did not allow the bumblebees to produce offspring. This study will be helpful in order to develop nutritive pollen diets for bumblebees.

**Abstract:**

Pollen is a major source of proteins and lipids for bumblebees. The nutritional content of pollen may differ from source plants, ultimately affecting colony development. This study investigated the foraging preferences of *Bombus terrestris* in regard to four pollen species, i.e., oilseed rape, wild apricot, sunflower, and buckwheat, under laboratory conditions. The results show that *B*. *terrestris* diversified their preference for pollens; the bumblebees mostly preferred wild apricot pollen, whereas sunflower pollen was the least preferred. The colonies fed on a mixed four-pollen diet, with a protein–lipid ratio of 4.55–4.86, exhibited better development in terms of the number of offspring, individual body size and colony weight. The colonies fed with buckwheat and sunflower pollens produced a significantly lower number of workers and failed to produce queen and male offspring. Moreover, wild apricot pollen had the richest protein content (23.9 g/100 g) of the four pollen species, whereas oilseed rape pollen had the highest lipid content (6.7 g/100 g), as revealed by the P:L ratios of wild apricot, sunflower, buckwheat, and oilseed rape, which were 6.76, 5.52, 3.50, and 3.37, respectively. Generally, *B*. *terrestris* showed feeding preferences regarding different pollens and a mixture of pollens, which ultimately resulted in differences in colony development. The findings of this study provide important baseline information to researchers and developers of nutritive pollen diets for bumblebees.

## 1. Introduction

Bumblebees (Apidae, *Bombus*) are important pollinators for many wildflowers and crops [1,2]. There are about 250 bumblebee species found worldwide, including in the Arctic and in temperate and tropical regions [3,4,5,6,7]. Under greenhouse conditions, bumblebees are the most reliable pollinators, especially for Solanaceae crops, such as tomatoes, eggplants and chilis. This is attributed to their characteristics, such as buzz pollination behavior; larger body with more hair; and tolerance to abiotic conditions, like a low temperature and low light [8,9].

In nature, bumblebees produce one generation a year, with bees collecting pollen and nectar from flowers for their food [10]. Pollen is a major source of proteins and lipids for colony development; additionally, pollen has trace levels of vitamins, secondary metabolites, carbohydrates, and the vital elements needed for bee larval growth [11,12,13]. Studies have shown that protein content affects the reproduction, physiological function, immunity, and larval development of bees [14,15,16,17,18,19]. Lipids play a vital role in bee development, as the metabolism of lipids is involved in various life activities of bees, such as growth and development [20], molting hormone production [21], learning ability improvement [22], gland formation [21] and brooding work [23]. Although pollen constitutes a nutritional resource for bumblebees as an essential source of proteins and lipids, the content and proportion of these nutrients differ between source plants. The content of proteins and lipids in pollen ranges between 2% and 60% and 2% and 20%, respectively [24]. Previous studies have shown that bumblebees prefer pollen that is rich in protein [25,26,27,28]. Vaudo and Stabler found that, in a single synthetic diet, 14:1 and 12:1 were the protein–lipid ratios (P:L ratios) of choice for *Bombus terrestris* and *B*. *impatiens* [29]. A semi-field experiment revealed that the plants selected had the highest P:L ratio (4.6:1) and that the feeding of *B. impatiens* changed with the P:L ratio. A study aiming to enhance pollen nutrition found that certain collecting bees of *B. impatiens* prefer to visit flowers with a P:L ratio between 5:1 and 10:1 [30].

*Bombus terrestris* is widely used commercially as an efficient pollinator of many crops, especially the crops in greenhouses. However, rearing of *B*. *terrestris* is familiar to only few companies, which keep the mass rearing techniques as business secrets [8]. Little is known to the public about how to raise bumblebees on a large commercial scale, for instance, in China, which is the largest market with high agricultural pollination demand. Food is an important factor that affects the reproductive success of effectiveness of bumblebee breeding [31]. This study used four kinds of pollens, i.e., oilseed rape, wild apricot, sunflower, and buckwheat, to assess the feeding preferences of *B. terrestris* under freely foraging conditions. Feeding preferences for the four kinds of pollens during the bumblebee colony development and their ultimate effects on the number of bees, body weight, and size were examined in terms of the development of bee individuals and colonies. This study was designed to optimize the artificial diet of breeding bumblebees and to provide a reference for improving the characteristics of bee colonies and breeding efficiency.

## 2. Materials and Methods

### 2.1. Pollens and Bumblebees

Four commercially available pollens of the local plants were used to feed bumblebees in this study (Appendix A). The commercial pollens were taken from honeybees in 2022. Each of the pollens—wild apricot, oilseed rape, buckwheat, and sunflower pollens—were collected fresh from three major producing regions of China (Appendix A) and stored in a refrigerator at –20 °C. These pollens were later transported to the Beijing laboratory on dry ice in a cold chain for experimental use. Each of pollen species was evenly taken from three locations (Appendix A) to feed the colonies, in order to minimize variation.

*Bombus terrestris* were purchased from the company Woofuntech Biocontrol (Hengshui, China). The bumblebees were reared in a laboratory room (28 ± 1 °C, 50% ± 5% RH) corresponding to Zhao’s study [32] at the Institute of Apicultural Research, Chinese Academy of Agricultural Sciences, Beijing. A total of 26 small bumblebee colonies were prepared, and each colony contained one queen and 10 workers.

### 2.2. Different Pollen Diets Used to Feed Bumblebees

#### 2.2.1. Testing Food Preference during Colony Development of Bumblebees with Four Different Pollen Varieties under Laboratory Conditions

A total of 10 bumblebee colonies were used for the food preference study. Each beehive was placed in a transparent plastic box (40 cm × 30 cm × 18 cm), and a 2 cm hole was cut in the front of each plastic box to allow for the free movement of the bumblebees. The same amounts of oilseed rape, wild apricot, sunflower, and buckwheat pollens were placed on plastic plates with the same distances between them near the hive entrance (Figure 1). The ten colonies were divided into four groups with pollens placed in different spatial orders to reduce the influence of pollen placement order on feeding outcomes (Table 1). This was to allow the bumblebees to forage freely without too much manual intervention. Finally, four colonies, with one best colony that was the most productive selected from each group, were used for preference analyses.

The experiment was conducted for 48 consecutive days between October and December 2022. Four kinds of pollen of the same weight were presented to the bumblebees. The amount of pollen consumed was determined and analyzed the next day between 10:30 and 11:30 a.m. The feeding preferences of the bumblebees for the four pollen species were recorded during the experiment, and it was observed whether this preference changed with colony development. An electronic balance and vernier calipers were used to measure the weight and left forewing length of the bumblebees (including queens, males, and workers). The experiment was conducted from feeding a small colony with 10 workers to the end of new queens emerging as adults. During the experiment, 100 newborn queens, males, and workers were taken from the best four colonies for weighing and measurement of the wing length. The weights of the hives were measured, and the numbers of workers and queens were recorded when the experiment finished.

#### 2.2.2. Monitoring Bumblebee Colony Development When Feeding a Single Pollen Type under Laboratory Conditions

At the same time, 16 bumblebee colonies were divided into four groups for a single-pollen study, and they were provided with oilseed rape, wild apricot, sunflower, and buckwheat pollens. Each single pollen study had four replicates. Sucrose solution and pollen diets were changed every day between 10:30 and 11:30 a.m. During the feeding process, an electronic balance and vernier calipers were used to measure the weight and left forewing length of the bumblebees (including queens, males, and workers). In the experiment, feeding continued until queens emerged as adults. Additionally, during the experiment, 100 newborn queens, males, and workers were taken from each single pollen study for weighing and measuring the wing length. Weights of the hives were also measured, and the numbers of workers and queens were recorded when the experiment finished.

### 2.3. Determination of Pollen Proteins and Lipids

#### 2.3.1. Determination of Protein Content of the Four Pollens

In the experiment, each type of pollen was repeated three times. Kjeldahl analysis following He [33] was used to determine the protein content of the four kinds of pollens. After weighing 1.00 g of pollen into a digestion tube with 0.5 g of CuSO_4_ and 4.5 g of K_2_SO_4_, 10 mL of concentrated sulfuric acid was slowly added and mixed via shaking. The digestion tube was placed into a high-temperature digestion furnace. The device was connected and the digestion tube was set to 240 °C pre-digestion for 30 min. It was adjusted to 430 °C high-temperature digestion for 1.5 h until the digestive solution was transparent blue-green. Then it was heated additionally for 20 min. After the digestive solution cooled down, the digestion tube was placed into a Kjeldahl nitrogen analyzer to complete distillation and titration. The mass of nitrogen was calculated and multiplied by 6.25 (the average coefficient for nitrogen conversion to crude protein) to obtain the crude protein content of pollen.

#### 2.3.2. Determination of Lipid Content of the Four Pollens

In the experiment, each type of pollen was repeated three times. The pollen was ground with a mortar and pestle, and 2 g of pulverized pollen was measured. It was placed on a filter paper and dried in an oven at 36 °C for 24 h until the weight remained constant, and then it was weighed again. The filter paper pack was placed in a dry leaching tube, and cotton was inserted into the upper inlet of the condenser tube. The lower half of the leaching tube was connected to an extraction flask, and about 1/2 of petroleum ether was added to the extraction flask. The extraction vials were placed in a thermostatic water bath, the cooling water was turned on, and the temperature of the heated water bath was maintained at 80 °C so that the petroleum ether condensed and dripped back once a minute. The procedure was carried out over a period of 6 h, with the petroleum ether being siphoned off 6–8 times per hour until it dripped onto the filter paper leaving no residue. At the end of the extraction, the upper condenser tube and filter paper pack were removed. The condenser was refitted, and the petroleum ether was recovered via distillation in a water bath. The filter paper pack was dried in an oven at 100 °C to dry the solvent, and then it was accurately weighed. The crude lipid content was calculated using the following formula [34]:Crude lipid% = (W1 − W2)/W × 100%

W1—filter paper package weight before suction filtration;

W2—filter paper package weight after suction filtration;

W—sample weight.

#### 2.3.3. Protein Consumption, Lipid Consumption, and P:L Ratio

The protein consumption and lipid consumption were calculated using the following formulas:Protein consumption% = MP/(MP + ML) × 100%
Lipid consumption% = ML/(MP + ML) × 100%

MP—mass of protein;

ML—mass of lipid.

The protein content was divided by the lipid content to derive the P:L ratio [30].

### 2.4. Data Analysis

To analyze the data, IBM SPSS 26 (Chicago, IL, USA) software was used [28]. Shapiro–Wilk tests were used to determine normality and Levene tests were used to test homogeneity of variance before any test. To determine whether the bee colony’s preference for various pollen changed over different stages, repeated measures ANOVA analysis was used, with the consumption of different pollen as the response variable, different time points as the within-subject factor, and pollen diet as the between-subject factor. Mauchly’s sphericity test showed that the hypothesis of sphericity was violated (*p* < 0.001); Therefore, Greenhouse–Geisser correction was used (ε = 0.363).

The data of body weight and wing length of bumblebees (including queens, males, and workers) from both preference and no-preference experiments presented significant differences in variance (Levene test, *p* < 0.001), and the nonparametric Kruskal–Wallis method was, therefore, used to test whether different pollen diets affect bumblebee individual’s weight and size, with the body weight and wing length of queens, males, and workers as response variables and pollen diet as the fixed factor.

To determine whether different pollen diets affect bumblebee colony development, one-way ANOVAs were used, with the number of workers, the number of queens, and the hive weight as response variables and pollen diet as the fixed factor. The colony was used as the replicate.

To compare nutrient contents among different pollen diets, one-way ANOVA was used, with protein content, fat content, and P:L ratio as response variables and pollen type as the fixed factor. The pollen was used as the replicate.

To analyze the changes in protein and lipid consumption patterns as well as the P:L ratio over time, repeated measures ANOVA analysis was used, with the proportion of consumed proteins, lipids, and P:L from the preference experiment as the response variables, the colony as the replicate, and different time points as the within-subject factor. Mauchly’s sphericity test showed that the hypothesis of sphericity was violated (*p* < 0.001); therefore, Greenhouse–Geisser correction was used (ε = 0.343).

## 3. Results

### 3.1. Bumblebees Prefer Wild Apricot Pollen during Colony Development

The bumblebee populations bred in the same environment for six weeks displayed a marked preferences towards the test pollens. The bumblebees were found to significantly prefer wild apricot pollen over the other pollens during colony development (F = 371.906, df = 3.158, *p* < 0.01), but they had comparable preferences for the pollens of buckwheat and oilseed rape. Interestingly, sunflower pollen was liked the least (Figure 2). Additionally, it was found that there was no significant difference in the proportion of the four pollens consumed at different times of bumblebee colony development (F = 0, df = 1.815, *p* > 0.05) (Figure 3). Among them, the mean ratios of the four pollens preferred by bumblebees were calculated to be 0.43: 0.23: 0.23: 0.11 (wild apricot: oilseed rape: buckwheat: sunflower).

### 3.2. Effects of Different Pollen Diets on Bumblebee Colony Development

#### 3.2.1. Effects on Bumblebee Individuals

The queens fed mixed pollen had a significantly higher weight (0.780 ± 0.121 g) than those fed either wild apricot (0.663 ± 0.086 g) (*p* < 0.01) or oilseed rape pollen (0.653 ± 0.096 g) (*p* < 0.01). However, there was no significant difference in the weight of the male and worker bees between the two types of pollen. No queens or males were produced in the colony fed buckwheat or sunflower pollen. Additionally, the workers fed mixed pollen had a significantly higher weight (0.267 ± 0.065 g) than those fed with buckwheat (0.177 ± 0.056 g) (*p* < 0.01) or sunflower pollen (0.131 ± 0.045 g) (*p* < 0.01) (Figure 4A).

The wing length of the queens fed mixed pollen (17.29 ± 0.56 mm) did not differ significantly from that of the queens in the group fed wild apricot pollen (16.99 ± 0.61 mm) (*p* > 0.05), but a significant difference (*p* < 0.01) was observed with the queens in the group fed oilseed rape pollen (17.15 ± 0.47 mm). No statistically significant variation (*p* > 0.05) was observed among the wing lengths of the males in the mixed-pollen (14.25 ± 0.23 mm), oilseed rape pollen (14.10 ± 0.54 mm), and wild apricot pollen (14.14 ± 0.23 mm) groups. The wing length of the workers fed mixed pollen was significantly longer than that of the workers in all the other groups (*p <* 0.05) (Figure 4B).

#### 3.2.2. Effects on Bumblebee Colonies

The number of worker bees in the mixed-pollen group (329 ± 37 bees) was not significantly different from that in the wild apricot (320 ± 42 bees, F = 39.990, df = 4, *p* > 0.05) and oilseed rape (314 ± 44 bees, F = 39.990, df = 4, *p* > 0.05) groups, but it was significantly different from that of the buckwheat (121 ± 28 bees) and sunflower (113 ± 21 bees) groups (F = 39.990, df = 4, *p* < 0.01) (Figure 5A). There was no significant difference in the number of queens between the mixed group (71 ± 4 queens) and the wild apricot group (62 ± 9 queens) (F = 2.713, df = 2, *p* > 0.05), but there was a higher significant difference with the oilseed rape group (57 ± 11 queens) (F = 2.713, df = 2, *p* < 0.01), while there were no queens produced in the buckwheat or sunflower groups (Figure 5B). The weight of the hives of the mixed-pollen group (0.282 ± 0.022 kg) was significantly higher than that of the other groups (F = 49.186, df = 4, *p* < 0.05). There were no significant differences between the wild apricot (0.235 ± 0.045 kg) and oilseed rape groups (0.210 ± 0.030 kg) (F = 49.186, df = 4, *p* > 0.05), and there were no significant differences between the buckwheat (0.090 ± 0.016 kg) and sunflower groups (0.058 ± 0.015 kg) (F = 49.186, df = 4, *p* > 0.05) (Figure 5C).

### 3.3. Protein and Lipid Consumption of Bumblebees during Colony Development

#### 3.3.1. Protein and Lipid Contents and P:L Ratio of the Four Pollens

All protein contents were significantly different (F = 226.154, df = 3, *p* < 0.01), with wild apricot pollen having the highest crude protein content (23. 9 ± 0.31 g/100 g), followed by oilseed rape pollen (22.3 ± 0.47 g/100 g), sunflower pollen (19.4 ± 0.52 g/100 g), and buckwheat pollen (15.7 ± 0.30 g/100 g) (Figure 6A). The lipid content of wild apricot (3.6 ± 0.45 g/100 g) was not significantly different (F = 34.572, df = 3, *p* > 0.05) from that of sunflower (3.5 ± 0.32 g/100 g), and it was highly significantly different (F = 34.572, df = 3, *p* < 0.01) from that of oilseed rape (6.7 ± 0.45 g/100 g) and buckwheat (4.5 ± 0.26 g/100 g) (Figure 6B). Regarding the P:L ratio, it was found that wild apricot pollen had the highest P:L ratio, followed by sunflower and buckwheat pollen, whereas rapeseed pollen had the lowest P:L ratio (Figure 6C).

#### 3.3.2. Consumption of Proteins and Lipids, and P:L Ratios of Bumblebees during Colony Development

The percentages of protein (F = 4.915, df = 5.000, *p* > 0.05) (Figure 7A) and lipid (F = 3.467, df = 1.708, *p* > 0.05) (Figure 7B) consumed did not differ significantly over the six weeks. Although the P:L ratios decreased in the second week (4.55 ± 0.53), they generally increased over the weeks (F = 6.459, df = 1.713, *p* < 0.05) (Figure 7C).

## 4. Discussion

Some bumblebee species have a wide range of diets, whereas others have an extreme preference for particular food plants [35]. In the artificial rearing of bumblebees, species depending on a wide range of diets can be reared more easily than those preferring specialized diets [5]. The pollen collected by bumblebees serves as a major source of the lipids and proteins that promote colony formation [2]. Nonetheless, the nutrient content of pollen varies depending on the source plant, which may result in different feeding preferences and have different impacts on the colony development of bumblebees [15,36,37].

The results of earlier studies [25,26,27,28] are in line with our data (Figure 2 and Figure 6), suggesting a preference for pollen with a high protein content. Amino acids, in addition to proteins and the P:L ratio, may also help to explain bumblebee foraging behavior. Wild apricot pollen has been found to have large quantities of proline in previous research [38]. Based on past research on bees, it is been suggested that proline may be the reason why the bumblebees in this experiment preferred wild apricot pollen [38,39]. Furthermore, research has demonstrated that *B*. *impatiens* favors a pollen diet with a greater P:L ratio when there are no other floral cues present [30]. In this study, it was also revealed that, due to its higher P:L ratio, *B. terrestris* prefers wild apricot pollen over other pollen species; the P:L ratio was 4.55–4.86 (Figure 7C), and this is a better ratio for raising the colony. However, other studies have shown that pollen with ratios of 5:1 and 10:1 is the most attractive to the workers of *B. impatiens* [30]. This may be due to different species having varying demands for proteins and lipids.

There are various reasons for preferences, including color, odor, and location, among which nutrition is only part of them. However, the purpose of this experiment was to explore preferences and improve artificial feeding, so the nutritional aspect was still the epicenter. In the experiment, the bumblebees exhibited remarkable intelligence in selecting the nutrients required from various pollens. The protein contents of oilseed rape pollen and buckwheat pollen differed significantly, but the bumblebees showed similar foraging preferences for them (Figure 2 and Figure 3); the reason for this is unknown. Research indicates that honeybees’ preferences for foraging are influenced by the amino acid composition [40,41], lipids [42,43], phenolics [43], protein levels [26,28,44], and P:L ratio [30] of pollens. More research is needed to identify the content and types of amino acids, fatty acids, and sterols in each pollen, to explore the relationships between nutrient components and bumblebee preferences, and to determine the factors that influence preferences. It is necessary to analyze more precise nutritional indicators of the four types of pollen and to verify them in bumblebee colonies in the future.

The mixed-pollen diet showed better results than the single-pollen diets in *B. terrestris* (Figure 4 and Figure 5; Appendix A). Additionally, the average optimally foraged pollen ratios were 0.43:0.23:0.23:0.11 for wild apricot, oilseed rape, buckwheat, and sunflower, respectively, over the period of six weeks (Figure 3). This may be attributed to variations in the source plants of pollen species, which compensated for the nutritional imbalance caused by single-pollen diets, thus providing the bumblebees with comprehensive nutrition [14,31,45]. As an illustration of poor reproductive success, the bumblebee colonies exposed to buckwheat and sunflower pollens produced fewer, smaller-sized workers, in addition to lacking males and queens (Figure 4 and Figure 5). This may also be explained by certain pollens having deficiencies in nutrients. Furthermore, a low protein content [46,47], the absence of three essential amino acids [46], the presence of alkaloids [48], and pollen wall spines that function as digestive barriers to prevent nutrient assimilation [49] might be associated with poor reproductive success in the colonies fed sunflower pollen.

## 5. Conclusions

Pollen is an important source of amino acids, minerals, vitamins, and lipids for bumblebees. Bumblebees gather pollen using specific cues as a guide. Studying the preferences of managed *Bombus terrestris* for various pollens will help in increasing colony development and reproductive efficiency, which can enhance conservation efforts and reduce the pollination deficit. Wild apricot pollen was the most favored by the bumblebees in this study, followed by oilseed rape, buckwheat, and sunflower pollens. This can be explained by variations in the pollens’ P:L ratio and protein concentrations. Successful colony formation was demonstrated by the bumblebees that were fed wild apricot, oilseed rape, buckwheat, and sunflower, which had proportions of 0.43, 0.23, 0.23, and 0.11 in their mixed-pollen diets, respectively. Diets devoid of certain critical nutrients, such as sunflower and buckwheat pollen diets, may have contributed to the reproductive failure of the *B. terrestris* colonies. The findings of this study can help in optimizing artificial rations for bumblebees in the future.

## Figures and Tables

**Figure 1 insects-15-00285-f001:**
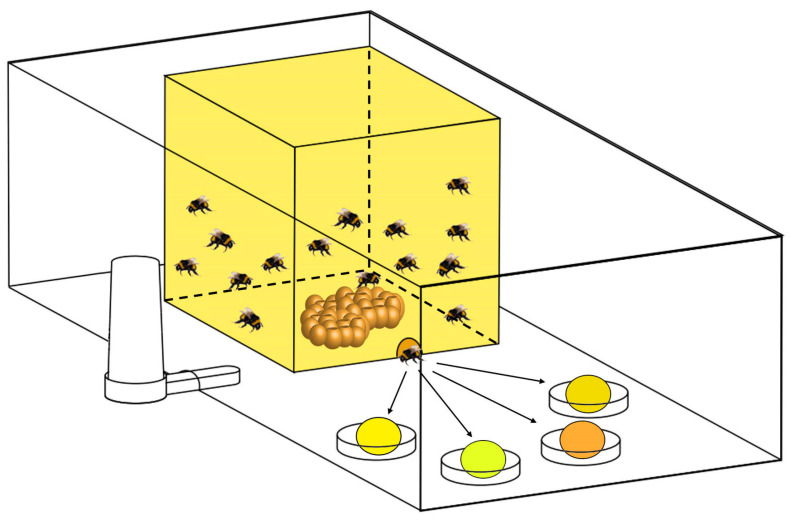
A graphical display of bumblebees’ feeding preference with the mixed four-pollen diet.

**Figure 2 insects-15-00285-f002:**
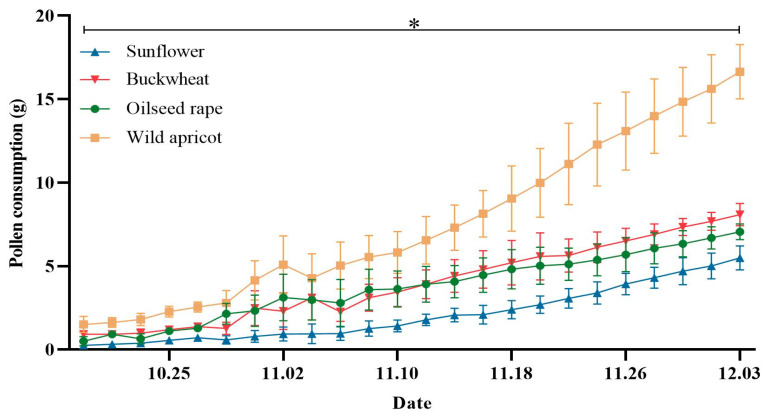
Pollen consumption of *Bombus terrestris* during colony development when fed four pollen diets freely under laboratory conditions. The pollen consumption data were obtained from four colonies. Data are presented as the mean ± S.D. * indicates a significant difference among the consumption of the four pollens.

**Figure 3 insects-15-00285-f003:**
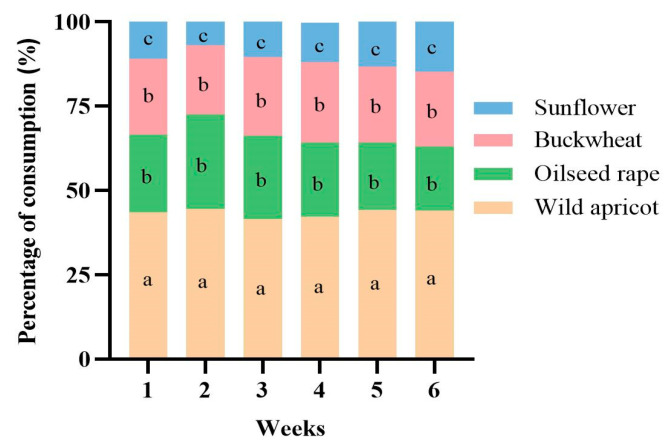
Percentage of the consumption of the four pollens by *Bombus terrestris* during colony development under laboratory conditions. The pollen consumption data were obtained from four colonies. Different letters indicate significant differences in a, b, c based on Kruskal–Wallis test at α = 0.05.

**Figure 4 insects-15-00285-f004:**
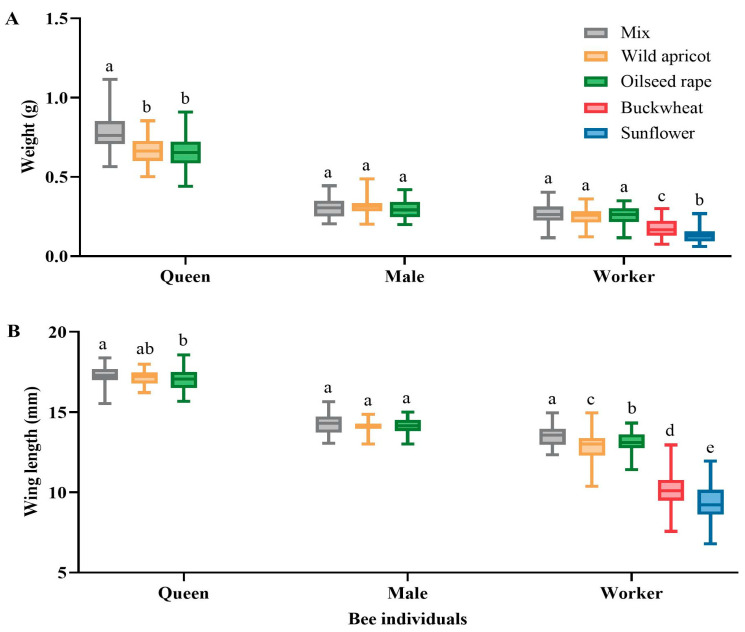
Effects of the five pollen diets on *Bombus terrestris* individuals. (**A**) Newborn weight of bumblebee individuals; (**B**) wing length of bumblebee individuals. The data on bee individuals were obtained from 100 repetitions of the five diets. Data are presented as the mean ± S.D. Different letters indicate significant differences in a, b, c, d based on Kruskal–Wallis test at α = 0.05.

**Figure 5 insects-15-00285-f005:**
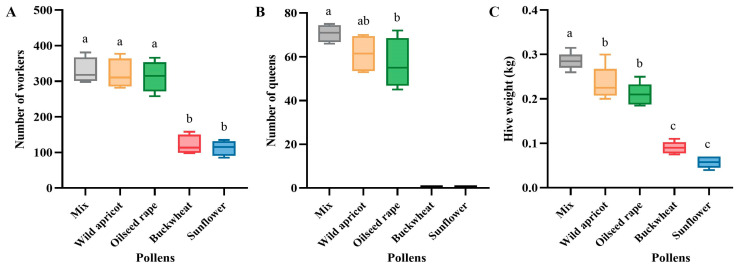
Effects of five pollen diets on *Bombus terrestris* colony (LSD, α = 0.05). (**A**) Number of workers; (**B**) number of queens; (**C**) hive weight. The data on bumblebee colonies were obtained from four colonies for each diet. Different letters indicate significant differences in a, b, c based on LSD at α = 0.05.

**Figure 6 insects-15-00285-f006:**
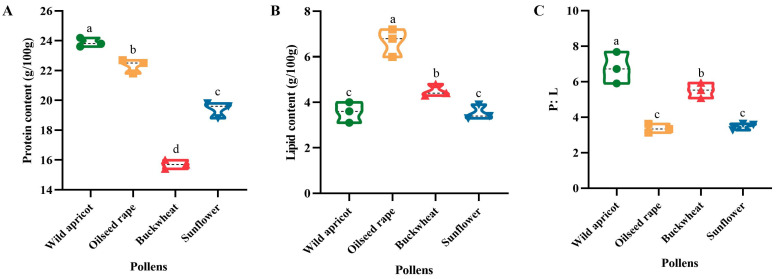
Nutritional contents of the four pollens in this study. (**A**) Protein content; (**B**) lipid content; (**C**) P:L ratios. The pollen nutrition data were obtained from three repetitions of four pollens. Different letters indicate significant differences in a, b, c, d based on LSD at α = 0.05.

**Figure 7 insects-15-00285-f007:**
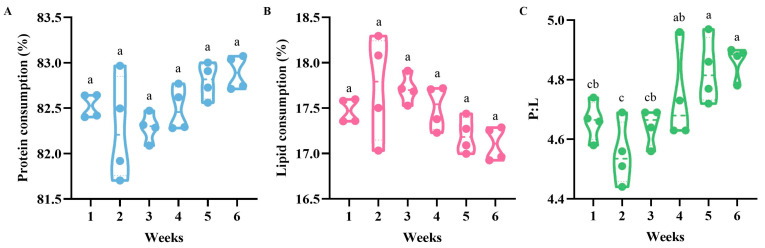
Consumption of proteins and lipids, and P:L ratios of *Bombus terrestris* during colony development when fed the four pollen diets. (**A**) Protein consumption in different weeks; (**B**) lipid consumption in different weeks; (**C**) P:L ratio in different weeks. The nutrition data were obtained over 6 weeks. Different letters indicate significant differences in a, b, c based on LSD at α = 0.05.

**Table 1 insects-15-00285-t001:** Pollen orders of *Bombus terrestris* food preference.

Pollen Spatial Arrangement (from Left to Right)	First	Second	Third	Forth
Colonies 1–3	Oilseed rape	Wild apricot	Sunflower	Buckwheat
Colonies 4–6	Buckwheat	Sunflower	Wild apricot	Oilseed rape
Colonies 7–8	Wild apricot	Oilseed rape	Buckwheat	Sunflower
Colonies 9–10	Sunflower	Buckwheat	Oilseed rape	Wild apricot

## Data Availability

All the data generated or analyzed during this study are included in this published article and its Appendix A.

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
