# Peer review of "Bombus terrestris* Prefer Mixed-Pollen Diets for a Better Colony Performance: A Laboratory Study"

_insects, 2024, doi:10.3390/insects15040285_

Round 1
Reviewer 1 Report
Comments and Suggestions for Authors
Very interesting work, well written and well presented. I have two questions about the manuscript.
1. Is there a reason why you choose these specific species of pollen in your experiments?
2. Do you know the amino acid synthesis of the proteins found in pollen, and respectively the type of lipids found in each pollen? How can each component affect bumblebee colonies?
If it is your next project it would be useful to mention it in the article.

Well written and well documented, some minor mistakes are included in the pdf file and a rephrase at lines 47-52, which must became more understandable.
Author Response
Response to the Reviewer 1 of
insects
for the manuscript insects-2813841
Bombus terrestris Prefer Mixed-Pollen Diets for a Better Colony Performance: A Laboratory Study
Reviewer 1 comments:
Very interesting work, well written and well presented. I have two questions about the manuscript.
>>> Many thanks for your comments, which have been very important in improving our manuscript.
Is there a reason why you choose these specific species of pollen in your experiments?
>>> The four species of pollen we have selected first is ecological relevance to our study. These are the common flowering plants in northern China where Bombus terrestris distributed in nature in China. Usually, local pollen sources can enhance the foraging behavior of bumblebees. Availability and feasibility were also encouraged to choose these pollens. These four pollens are commercially available in market and will be feasible in future use. These are two major reasons we chose these pollens. We have modified the description of pollen sources at the part of Materials and Methods. (please see the revised version lines 86-87).
Do you know the amino acid synthesis of the proteins found in pollen, and respectively the type of lipids found in each pollen? How can each component affect bumblebee colonies? If it is your next project it would be useful to mention it in the article.
>>> Yes, we just compared content of crude proteins and lipids between the four pollens in this study. But in the subsequent experiment, we will use HPLC and GC-MS methods to determine the content and types of amino acids, fatty acids, and sterols in each pollen, in order to further explore the relationships between the pollen nutrient components and the bumblebee preferences, and try to reveal what’s the functional components and how they affect bumblebee colony development in future. We added the relative sentences at the part of the Discusstion. (please see the revised version lines 324-328).

Reviewer 2 Report
Comments and Suggestions for Authors
The article is clear and easy to understand, and the experimental results have important implications for production applications.
Figure 3 and 4B Since the queen and male do not produced, the missing delete can be omit for statistics. It is not necessary to show their values in these graph.
Line 242 243 Check the spelling of F value, I am not aware the correct value was 34,572 or 34.572.
Author Response
Response to the Reviewer 2 of
insects
for the manuscript insects-2813841
Bombus terrestris Prefer Mixed-Pollen Diets for a Better Colony Performance: A Laboratory Study
Reviewer 2 comments:
The article is clear and easy to understand, and the experimental results have important implications for production applications.
>>> Many thanks for your comments, which have been very important in improving our manuscript.
Figure 3 and 4B Since the queen and male do not produced, the missing delete can be omit for statistics. It is not necessary to show their values in these graph.
>>> Many thanks again. We have updated the figure 3 and Figure 4B, and the related descriptions of analysis in part of the Result. (Please see the revised version lines 241, 253-254, and 261).
Line 242 243 Check the spelling of F value, I am not aware the correct value was 34,572 or 34.572.
>>> We checked the spelling of the F value and found the numerical mistakes. The correct value is 34.572. (please see the revised version lines 272-273).

Reviewer 3 Report
Comments and Suggestions for Authors
This study tests the ability of 5 diets to support colony growth in Bombus terrestris: 4 single pollen diets and 1 mixed diet. Similar to other studies the researchers find some diets better than others, especially those with higher protein concentration and, to some extent a higher protein to lipid ratio. The basic experimental approach seems well conceived and executed and, with some adjustments, will make a good publication.
There is one major problem with the manuscript as it is currently. The statistics are insufficiently described and, to the extent they are described, do not seem appropriate to the way the study was carried out. For no experiments are the statistical parameters fully described: what was the dependent variable and what were all the experimental factors? For the preference experiment, the analysis is given as a 2 way ANOVA with repeated measures. But what is the dependent variable? The bees ate different amounts of 4 different diets. Thus there are 4 dependent variables. This sounds like a MANOVA design. I don't understand what the analysis was. For bee body size, the statistical test is given as a Kruskal-Wallis test. The problem with a Kruskal-Wallis test is that it would have to ignore colony as a factor, and that should not be done. I assume diet alone was used as a factor in the test. For colony production metrics, the one-way ANOVA is probably correct but the authors should still describe the test more. I assume what was done was they used the total number of workers/males/queens as the dependent variable, diet as the factor and colony as a replicate. For the last analyses on protein and lipid ratios over time as a one-way ANOVA, it is unclear what went into the analysis. Did they ignore the effect of colony? Colony is a basic component of their experiment.
The rest of my comments go in order of the manuscript and are much narrower in focus.
Line 32 (Abstract): The 81% protein and 16.9% lipid values here do not mean anything to the reader. They are not standard measurements and are confusing, especially when the protein and lipid values of the diets are given afterward. The bees did not feed on an 81% protein diet; they fed on a diet in which the protein percentage of the protein + lipid components was 81%. That is very different. It also isn't needed in the abstract.
Line 74: Bombus terrestris is not known to be the most efficient pollinator worldwide. Such a study does not exist. It is widely used commercially as an efficient pollinator of many crops. Also, there actually is a lot known about raising bumblebees commercially.
Line 79: What developmental stages do you have in mind?
Line 86: Was the commercial pollen taken from honeybees (and thus having nectar mixed in with it) or hand collected?
Line 91: What does it mean that the pollen from different locations had the same weight? If you mean that when given in experimental treatments, the same amount was given of each pollen type then that should be stated in the experimental protocol not here.
Line 105: What are the four groups? Does this refer to the experimental bee colonies being in groups and getting a particular diet array throughout the experiment?
Line 109: I think you mean "presented to" instead of "fed to" since the bees are making a choice.
Line 115: How long was the experiment
Line 123-124: Unclear how often measurements of individual bees were taken. Were all bees measured every day? Or were they measured once and marked and not measured again? Were only the foragers measured or all bees in the colony?
Line 124: Should say "emerged as adults" rather than "born" for insects.
Line 129: "The He" isn't a method but is an author. Should probably say used a "Kjeldahl analysis following He[29]". Also, this method does not estimate protein but rather nitrogen. How did the authors convert their nitrogen estimate into protein?
Line 175: I don't know what "markable inclination" means.
Line 180: "Stages". I don't think you are recognizing any stages here but just different times.
Figures 1 and 2. Both of these figures are in the "preference" part of the results. They indicate that the data come from four colonies, but according to line 100 there were 10 colonies in the preference experiment. Do the data come from only four of the ten? If so, why?
Line 196: The weight measurements of bees. Fresh weight of living bees is not a reliable estimate of their body size, as they can hold at least half their own weight in nectar in their body. You have size estimates based on wing size. I suggest you eliminate the use of fresh weight and only use wing size estimates for body size. While the data sets are fairly similar you have some differences in outcome.
Line 213-214: What does it mean that the data were obtained from 100 repetitions of the five diets? What bees were measured? Were some measured more than once?
Line 217: Where does the number of workers come from? Is this the number of workers in the colony at the end? Is it the number of workers produced over the whole experiment?
Line 255: Explain how these percentages were calculated. I assume it is mass of protein/(mass of protein + mass of lipid) * 100.
Line 271: Perhaps you mean "colony" instead of "population"
Line 272: Which earlier studies are you referring to?
Line 278: "impatiently" ???
Line 279: The experimental design does not test the reason for the preference of apricot. The result is consistent with it being P:L ratio. Why wasn't buckwheat preferred over oilseed rape, given its higher P:L ratio?
Line 281: Do you mean "queen offspring?" All the offspring here are the offspring of queens, most likely.
Line 298: How do you know the pollen ratios were optimally foraged? Perhaps they would have done better on a diet that was 80% apricot rather than 40%.
Line 320: Do you know that the buckwheat and sunflower diets are devoid of critical nutrients? They could be somewhat toxic instead.
Comments on the Quality of English LanguageThe English is quite good. I have indicated in my comments most of the places in which something is unclear.
Author Response
Response to the Reviewer 3 of
insects
for the manuscript insects-2813841
Bombus terrestris Prefer Mixed-Pollen Diets for a Better Colony Performance: A Laboratory Study
Reviewer 3 comments:
This study tests the ability of 5 diets to support colony growth in Bombus terrestris: 4 single pollen diets and 1 mixed diet. Similar to other studies the researchers find some diets better than others, especially those with higher protein concentration and, to some extent a higher protein to lipid ratio. The basic experimental approach seems well conceived and executed and, with some adjustments, will make a good publication.
>>> Many thanks for your insightful comments, which have been very significantly improve our manuscript.
There is one major problem with the manuscript as it is currently. The statistics are insufficiently described and, to the extent they are described, do not seem appropriate to the way the study was carried out. For no experiments are the statistical parameters fully described: what was the dependent variable and what were all the experimental factors? For the preference experiment, the analysis is given as a 2 way ANOVA with repeated measures. But what is the dependent variable? The bees ate different amounts of 4 different diets. Thus there are 4 dependent variables. This sounds like a MANOVA design. I don't understand what the analysis was. For bee body size, the statistical test is given as a Kruskal-Wallis test. The problem with a Kruskal-Wallis test is that it would have to ignore colony as a factor, and that should not be done. I assume diet alone was used as a factor in the test. For colony production metrics, the one-way ANOVA is probably correct but the authors should still describe the test more. I assume what was done was they used the total number of workers/males/queens as the dependent variable, diet as the factor and colony as a replicate. For the last analyses on protein and lipid ratios over time as a one-way ANOVA, it is unclear what went into the analysis. Did they ignore the effect of colony? Colony is a basic component of their experiment.
>>> We have extended the descriptions of statistics and listed the dependent variables and predictor factors in the revised version.
For preferences analysis, we considered pollen consumption as a dependent variable whereas time and diet as independent variables to run a two-way ANOVA (please see line 179-184 in the revised version).
For the body size, we used non parametric, Kruskal-Wallis test, after our data failed to meet the assumptions of a parametric one-way ANOVA. However, we only focused at the influence of diet on bumblebee body weight and size. In this experiment, the bumblebee colony was only used as a replicate. (please see line 185-189 in the revised version).
For colony production metrics, we used one-way ANOVAs to determine pollen diet influence on bumblebee colony development. Here, we considered the number of workers, the number of queens, and the hive weight as response variables and pollen diet as independent variable. (please see line 190-192 in the revised version).
For changes of protein and lipid ratios consumption over time, we conducted repeated measures ANOVA considering the proportion of consumed proteins, lipids, and P: L as response variables, and time as an independent variable. (please see line 196-200 in the revised version).
The rest of my comments go in order of the manuscript and are much narrower in focus.
Line 32 (Abstract): The 81% protein and 16.9% lipid values here do not mean anything to the reader. They are not standard measurements and are confusing, especially when the protein and lipid values of the diets are given afterward. The bees did not feed on an 81% protein diet; they fed on a diet in which the protein percentage of the protein + lipid components was 81%. That is very different. It also isn't needed in the abstract.
>>> We have deleted the sentence on this in the Abstract. (please see the revised version lines 31-32)
Line 74: Bombus terrestris is not known to be the most efficient pollinator worldwide. Such a study does not exist. It is widely used commercially as an efficient pollinator of many crops. Also, there actually is a lot known about raising bumblebees commercially.
>>>Yes, we have revised the descriptions.
Bombus terrestris is widely used commercially as an efficient pollinator of many crops, especially the crops in greenhouses. However, rearing of B. terrestris is familiar to only few companies, which keep the mass rearing techniques as business secrets. Little is known to the public about how to raise bumblebees on a large commercial scale, for instance, in China which is the largest market with high agricultural pollination demand. (please see the revised version lines 71-75).
Line 79: What developmental stages do you have in mind?
>>> We improved the sentence by replacing “… at different developmental stages …” with “… during the bumblebee colony development …” for clarity of intended meaning (please see the revised version lines 79-80). We also improved the descriptions of ‘different pollen diets used to feed bumblebees’ in the Methods part. (please see the revised version lines 105-109).
Line 86: Was the commercial pollen taken from honeybees (and thus having nectar mixed in with it) or hand collected?
>>> The commercial pollens were taken from honeybees by beekeepers in 2022, and we have supplemented it. (please see the revised version lines 86-87).
Line 91: What does it mean that the pollen from different locations had the same weight? If you mean that when given in experimental treatments, the same amount was given of each pollen type then that should be stated in the experimental protocol not here.
>>> The pollen we selected for this study came from three different regions. In order to mitigate variations in pollen nutritional content among different regions, we mixed the pollen from the three regions uniformly before conducting our experiments. (please see the revised version lines 91-92).
Line 105: What are the four groups? Does this refer to the experimental bee colonies being in groups and getting a particular diet array throughout the experiment?
>>> The four groups delineate distinct four arrangement methods, aimed at eliminating the influence of pollen placement order on feeding outcomes. (please see the revised version lines 105-110).
Line 109: I think you mean "presented to" instead of "fed to" since the bees are making a choice.
>>>We have modified it to “… presented to …” instead of "… fed to …". (please see the revised version line 112).
Line 115: How long was the experiment
>>> The experiment of preference study lasted for a total of 48 days from October 17, 2022 to December 3, 2022. (please see the revised version lines 111-112).
Line 123-124: Unclear how often measurements of individual bees were taken. Were all bees measured every day? Or were they measured once and marked and not measured again? Were only the foragers measured or all bees in the colony?
>>> We didn’t the measurements every day. We measured the newborn weight and forewing length of workers, males, and queens, a total of 100 individuals for each. We added the measurements for individuals, the weights of hives and the numbers of workers and queens during the experiment. (please see the revised version lines 118-122).
Line 124: Should say "emerged as adults" rather than "born" for insects
>>> We have improved the sentence by replacing “… queens were born …” with “… queens emerged as adults …”. (please see the revised version line 119 and 131).
Line 129: "The He" isn't a method but is an author. Should probably say used a "Kjeldahl analysis following He[29]". Also, this method does not estimate protein but rather nitrogen. How did the authors convert their nitrogen estimate into protein?
>>> We have revised the description.
“Kjeldahl analysis following He [29] was used to determine …” (please see the revised version lines 137).
Also, to obtain the crude protein content, the mass of nitrogen was measured and multiplied by 6.25 (the average coefficient of nitrogen conversion to crude protein). (please see the revised version lines 145-147).
Line 175: I don't know what "markable inclination" means.
>>> We have replaced “… markable inclination …” with “… markable preferences …”. (please see the revised version line 205).
Line 180: "Stages". I don't think you are recognizing any stages here but just different times
>>> Yes, we have replaced “… stages ...” with “… different times ...”. (please see the revised version line 210).
Figures 1 and 2. Both of these figures are in the "preference" part of the results. They indicate that the data come from four colonies, but according to line 100 there were 10 colonies in the preference experiment. Do the data come from only four of the ten? If so, why?
>>> Yes, the data of the figures 1 and 2 comes from four of the ten colonies in the preference experiment, because some of colonies did not develop well and some queens died during the experiment. So, we selected the best colony of each group as Table 1 for preference analyses. We improved the descriptions for this in part of the Methods. (please see the revised version line 120).
Line 196: The weight measurements of bees. Fresh weight of living bees is not a reliable estimate of their body size, as they can hold at least half their own weight in nectar in their body. You have size estimates based on wing size. I suggest you eliminate the use of fresh weight and only use wing size estimates for body size. While the data sets are fairly similar you have some differences in outcome.
>>> Yes, for bumblebees, wing length is more reliable than fresh weight for body size estimation, but newborn weight is a very common indicator too for bee size. In this regard, we would like to keep both in text. (please see the revised version lines 226-232).
Line 213-214: What does it mean that the data were obtained from 100 repetitions of the five diets? What bees were measured? Were some measured more than once?
>>> For each pollen group, 100 queens, 100 males and 100 workers were tested. Each bee is only measured once. We have selected newborn bumblebees for measurement. The fur color of newborn bumblebees is white and easy to identify. After measurement, it is placed back to the colony. (please see the updated revised version lines 119-220 and 131-132).
Line 217: Where does the number of workers come from? Is this the number of workers in the colony at the end? Is it the number of workers produced over the whole experiment?
>>>It comes from the number of workers generated throughout the experiment. (please see the updated revised version lines 118-119 and 130-131).
Line 255: Explain how these percentages were calculated. I assume it is mass of protein/(mass of protein + mass of lipid) * 100.
>>>Yes, percentages were calculated as – mass of protein/(mass of protein + mass of lipid) * 100%. For example, in the first week, the total amount of protein consumed by the bee colony was calculated to be 1308.55g, and the fat content was 280.285g, resulting in a protein content of 82.36% (Figure 6.A). (please see the updated revised version lines no. 169-174)
Line 271: Perhaps you mean "colony" instead of "population"
>>> Yes, we have replaced “population” with “colony” here. (please see the updated revised version line no. 301)
Line 272: Which earlier studies are you referring to?
>>> We have added the references. (please see the updated revised version line no. 302)
Line 278: "impatiently" ???
>>> We have removed the word 'impatiently '. (please see the updated revised version line no. 307-309)
Line 279: The experimental design does not test the reason for the preference of apricot. The result is consistent with it being P:L ratio. Why wasn't buckwheat preferred over oilseed rape, given its higher P:L ratio?
>>> This experiment can only roughly analyze the reasons for different pollen preferences from a nutritional perspective. But more specific reasons require further experimental research, such as detecting the amino acid, fatty acid, alkaloid content of pollen. (please see the updated revised version lines no. 324-328)
Line 281: Do you mean "queen offspring?" All the offspring here are the offspring of queens, most likely.
>>> The meaning of queen offspring is bee colony, and we have modified the description method. (please see the updated revised version line no. 311)
Line 298: How do you know the pollen ratios were optimally foraged? Perhaps they would have done better on a diet that was 80% apricot rather than 40%.
>>> This ratio was calculated based on the free feeding of bumblebees. In this regard, we consider these as the best ratios chosen by the bumblebees themselves. (please see the updated revised version lines no. 107-108) .
Line 320: Do you know that the buckwheat and sunflower diets are devoid of critical nutrients? They could be somewhat toxic instead.
>>> In subsequent experiments, we will determine that the total amount of amino acids and fatty acids in sunflowers and buckwheat is lower than that in wild apricots and rapeseed, and some specific components that affect growth and development, such as methionine, also show certain differences. (please see the updated revised version lines no. 324-328) . Toxins may also affect the development of bee colonies, and we will arrange relevant experiments to test the content and types of toxins in pollen in future.
Reviewer 4 Report
Comments and Suggestions for Authors
The topic of this manuscript is very interesting. Being able to establish which diet is optimal for bumblebees would have practical implications for the cultivation of those botanical species (for example in greenhouses) that use the help of bumblebees for pollination.
If I understand correctly, the thesis without pollen was not made. I personally would have added it considering that the weight, the length of the wings were measured and the bees were counted. this thesis would have been a control that in reality does not exist.
The article is well organized and well written.
Minor comment
· Page 2 lines 45-46, 53-54, 74-76 the bibliographical reference is missing. Please add it.
· Page 2 lines 66 and 68 I suggest first writing the full name "protein-lipid ratio" with the abbreviation in brackets and then always the abbreviation in the text.
· page 8 lines 268-269, 269-271 the bibliographical reference is missing. Please add it.
· Page 8 line 272 which previous studies are the authors referring to? Please clarify.
Comments on the Quality of English Languagethe article appears well written
Author Response
Response to the Reviewer 4 of
insects
for the manuscript insects-2813841
Bombus terrestris Prefer Mixed-Pollen Diets for a Better Colony Performance: A Laboratory Study
Reviewer 4 comments:
The topic of this manuscript is very interesting. Being able to establish which diet is optimal for bumblebees would have practical implications for the cultivation of those botanical species (for example in greenhouses) that use the help of bumblebees for pollination.
If I understand correctly, the thesis without pollen was not made. I personally would have added it considering that the weight, the length of the wings were measured and the bees were counted. this thesis would have been a control that in reality does not exist.
The article is well organized and well written.
>>> Many thanks for your comments, which have been very important in improving our manuscript.
Minor comment
Page 2 lines 45-46, 53-54, 74-76 the bibliographical reference is missing. Please add it.
>>> We have added the meticulously corresponding references. (please see the revised version lines 45-46, 52-53, and 76-77).
Page 2 lines 66 and 68 I suggest first writing the full name "protein-lipid ratio" with the abbreviation in brackets and then always the abbreviation in the text.
>>> We have modified with the full name "protein lipid ratio (P:L ratio)" in the first appearance, and then use the abbreviation in the whole text. (please see the revised version line 65 and the whole text).
page 8 lines 268-269, 269-271 the bibliographical reference is missing. Please add it.
>>> We have added the pertinent corresponding references. (please see the revised version lines 299-301).
Page 8 line 272 which previous studies are the authors referring to? Please clarify.
>>> We have added the pertinent corresponding references. (please see the revised version line 302)
New references added in the revised manuscript
- Gallai, N.; Salles, J.M.; Settele, J.; Vaissière, B.E. Economic valuation of the vulnerability of world agriculture confronted with pollinator decline. Ecological Economics 2009, 68, 810–821.
- Burkle, L.A.; Martin, J.C.; Knight, T.M. Plant-pollinator interactions over 120 years: loss of species, co-occurrence, and function. Science 2013, 339, 1611–1615.
- Friend W.G. Nutritional requirements of phytophagous insects. Annual Review of Entomology 1958, 3, 57–74.
- Moerman, R.; Vanderplanck, M.; Fournier, D.; Jacquemart, A.L.; Michez, D. Pollen nutrients better explain bumblebee colony development than pollen diversity. Insect conservation and diversity 2017, 10, 171–179.
- Moerman, R.; Roger, N.; Jonghe, R.D.; Michez, D.; Vanderplanck, M. Interspecific variation in bumblebee performance on pollen diet: new insights for mitigation strategies. PLoS ONE 2016, 11, e0168462.
Round 2
Reviewer 3 Report
Comments and Suggestions for Authors
The manuscript has been substantially improved since the first submission. The description of the experimental methodology is much clearer and I believe I understand the setup much better now. Although the description of the statistical approach has been improved, it is still incomplete and, in one case, problematic. I will go through the five paragraphs of the statistical description one by one.
Lines 177-184 focus on the preference experiment. The response variable is said to be consumption. Is it the amount of each food type consumed or is it, as indicated on lines 209-210, the proportion of total consumption attributable to each food type? Whether it is amount consumed or proportion, there is still a problem with using ANOVA to analyze these data. As I understand this experiment, the analysis was done using 4 colonies (the most productive ones in each food arrangement replicate) with 4 data points taken per time period per colony: the amount of each food type consumed (or the proportion of each food type consumed) at each time period for each colony. ANOVA includes the basic assumption of independence among the response variables. In the analysis described, I do not see how the data points taken for a single colony at a single time period can be considered independent. The colony needs a finite amount of food at one time. The more the colony takes of one kind of food, the less of another kind it needs. These are not independent of each other. If the dependent variable is proportion, then this is an even bigger problem because the proportions are constrained to add up to 1, and thus clearly not mathematically independent. This is a very well known statistical problem in choice studies involving more than 2 food types at a time, and there is a whole literature regarding how to handle these kinds of data. For example see,
Lockwood, JR. 1998. On the statistical analysis of multiple-choice feeding preference experiments. Oecologia 116:475-481.
Larrinaga, AR. 2010. A univariate analysis of variance design for multiple-choice feeding-preference experiments: A hypothetical example with fruit-eating birds. Acta Oecologica 36: 141-148.
These papers lay out the basic issues and some ways to resolve them, but the solutions are not simple. This is a difficult statistical problem caused by a simple experimental design, the one used in the current manuscript. In addition to the lack of independence among response variables, the papers also discuss the problem of change in food quality over time (autogenic change, i.e., drying of foods over time). I am less concerned with this problem (autogenic change) in the current dataset than with the lack of independence while using ANOVA to generate p-values. The authors need an analysis that does not suffer from a lack of independence among data points.
Lines 185-189 describe body weight and wing length analyses. What experiment does this description go with? The preference experiment, the no-preference consumption experiment, or both? The analysis, as described, does not include the factor of colony. Either colony must be a factor in this analysis (there is no reason to assume colony doesn't influence body size when there are multiple colonies getting the same treatment) or there can only be one measurement (i.e., an average) of body size per colony. Please include how colony is used in this analysis.
Lines 190-192 describe colony reproduction. I assume colony is used as the replicate here. If so, please state that in the description.
Lines 193-195 describe nutrient analysis. What constitutes a replicate in the analysis and how many replicates are there?
Lines 196-200 describe protein and lipid consumption patterns. Which experiment do these analyses go with? I assume it is the preference experiment, otherwise the ratios would be constant, but this needs to be stated. Is the colony the replicate here? Is that what "within-subject" refers to? Is colony a factor in the statistical model? Please state this in the description.
A few small corrections.
Line 109. Please describe what "best colony" means. Is it the most productive one?
Line 110. It would be better to say "pollen spatial arrangement" than "pollen order" because order can also mean order in time. In this case, all pollen is presented at one time.
Author Response
The manuscript has been substantially improved since the first submission. The description of the experimental methodology is much clearer and I believe I understand the setup much better now. Although the description of the statistical approach has been improved, it is still incomplete and, in one case, problematic. I will go through the five paragraphs of the statistical description one by one. >>> We feel great thanks for your professional review work on our article. As you are concerned, there are several problems that need to be addressed. According to your nice suggestions, we have made corrections to our previous draft, the detailed corrections are listed below. Lines 177-184 focus on the preference experiment. The response variable is said to be consumption. Is it the amount of each food type consumed or is it, as indicated on lines 209-210, the proportion of total consumption attributable to each food type? Whether it is amount consumed or proportion, there is still a problem with using ANOVA to analyze these data. As I understand this experiment, the analysis was done using 4 colonies (the most productive ones in each food arrangement replicate) with 4 data points taken per time period per colony: the amount of each food type consumed (or the proportion of each food type consumed) at each time period for each colony. ANOVA includes the basic assumption of independence among the response variables. In the analysis described, I do not see how the data points taken for a single colony at a single time period can be considered independent. The colony needs a finite amount of food at one time. The more the colony takes of one kind of food, the less of another kind it needs. These are not independent of each other. If the dependent variable is proportion, then this is an even bigger problem because the proportions are constrained to add up to 1, and thus clearly not mathematically independent. This is a very well known statistical problem in choice studies involving more than 2 food types at a time, and there is a whole literature regarding how to handle these kinds of data. For example see, Lockwood, JR. 1998. On the statistical analysis of multiple-choice feeding preference experiments. Oecologia 116:475-481. Larrinaga, AR. 2010. A univariate analysis of variance design for multiple-choice feeding-preference experiments: A hypothetical example with fruit-eating birds. Acta Oecologica 36: 141-148. These papers lay out the basic issues and some ways to resolve them, but the solutions are not simple. This is a difficult statistical problem caused by a simple experimental design, the one used in the current manuscript. In addition to the lack of independence among response variables, the papers also discuss the problem of change in food quality over time (autogenic change, i.e., drying of foods over time). I am less concerned with this problem (autogenic change) in the current dataset than with the lack of independence while using ANOVA to generate p-values. The authors need an analysis that does not suffer from a lack of independence among data points. >>>Because the consumption is calculated simply to obtain the consumption ratio, and the statistical methods are consistent with Two-way Repeated Measures ANOVA, the following description only uses ratios to explain. Two-way Repeated Measures ANOVA need to meet two requirements: 1. There is only one dependent variable and it is a continuous variable. 2. Each research object has two or more Within Subject Factors. In this experiment, the dependent variable is the proportion of consumption ratio, which satisfies the first requirement. Within subject factor is the number of weeks, with six levels, meeting the second requirement. Therefore, two-way repeated reasures ANOVA can be used. And different pollen belongs to the between subject factors, divided into four levels, corresponding to four types of pollen. The specific SPSS analysis chart is as follows(Please refer to the attachment for specific images. Images cannot be added here) In our statistics, taking the consumption ratio of oilseed rape pollen in the first week as an example, the four values (0.23, 0.24, 0.24, 0.21(in the first figure above)) were obtained from four groups of bee colonies, so in our statistics, different bee colonies are only replicates, not factors. Lines 185-189 describe body weight and wing length analyses. What experiment does this description go with? The preference experiment, the no-preference consumption experiment, or both? The analysis, as described, does not include the factor of colony. Either colony must be a factor in this analysis (there is no reason to assume colony doesn't influence body size when there are multiple colonies getting the same treatment) or there can only be one measurement (i.e., an average) of body size per colony. Please include how colony is used in this analysis. >>>The relationship between weight and wing length was statistically analyzed in both preference experiments and no-preference experiments. The preference experiment was considered as the mixed group which was freely consuming different pollen, while the no-preference experiment was the single pollen group, corresponding to the specific pollen groups consumed. In this analysis, different pollen is considered as the factor, and the bumblebees taken from different groups fed with the same pollen are repetition. The description has been modified, please see the revised version of line 187. Lines 190-192 describe colony reproduction. I assume colony is used as the replicate here. If so, please state that in the description. >>>Yes, the analysis group is duplicated, and the specific description has been changed. Please see the revised version of lines 196-197. Lines 193-195 describe nutrient analysis. What constitutes a replicate in the analysis and how many replicates are there? >>>The replicate is pollen. Each type of pollen has three sets of replicates. The description has been changed, please see the revised version of lines 138, 151 and 200. Lines 196-200 describe protein and lipid consumption patterns. Which experiment do these analyses go with? I assume it is the preference experiment, otherwise the ratios would be constant, but this needs to be stated. Is the colony the replicate here? Is that what "within-subject" refers to? Is colony a factor in the statistical model? Please state this in the description. >>>This experiment is related to preference experiments. Bee colonies are the replicate, not a factor. The description has been changed, please see the revised version of lines 203-204. A few small corrections. Line 109. Please describe what "best colony" means. Is it the most productive one? >>> The best bee colony refers to the group of bees with good growth status and no occurrence of queen bee death, which has the largest colony size. The description has been changed, please see the revised version of lines 109. Line 110. It would be better to say "pollen spatial arrangement" than "pollen order" because order can also mean order in time. In this case, all pollen is presented at one time. >>>We have replaced the "pollen order" to "pollen spatial arrangement" in the article. Please see revised version of lines 106 and 111.
